# *Vibrio* sp. and Identification of the *ctx* Gene of Cholera Toxin in the Mandinga Coastal Lagoon, Veracruz, Mexico

**DOI:** 10.3390/microorganisms13020352

**Published:** 2025-02-06

**Authors:** María del Refugio Castañeda-Chávez, Rosa Elena Aguilar-Muslera, Christian Reyes-Velázquez, Fabiola Lango-Reynoso, Rosa Elena Zamudio-Alemán, Magnolia Gricel Salcedo-Garduño

**Affiliations:** 1Tecnológico Nacional de México/Instituto Tecnológico de Boca del Río, Km 12 Carretera Veracruz-Córdoba, Boca del Río 94290, Mexico; mariacastaneda@bdelrio.tecnm.mx (M.d.R.C.-C.); christianreyes@bdelrio.tecnm.mx (C.R.-V.); fabiolalango@bdelrio.tecnm.mx (F.L.-R.); rosazamudio@bdelrio.tecnm.mx (R.E.Z.-A.); 2Tecnológico Nacional de México/Instituto Tecnológico de Veracruz, Miguel Ángel de Quevedo 2779, Veracruz 91897, Mexico; rosa.am@veracruz.tecnm.mx

**Keywords:** anthropogenic activity, pathogen, *Vibrio* sp., *Vibrio cholerae*

## Abstract

Coastal lagoons have undergone changes due to anthropogenic activities, the presence of wastewater discharges, and unsustainable practices that alter water quality, favoring the presence of pathogenic microorganisms such as *Vibrio*. This study identified the presence of the genes for zinc metalloproteinase (HA) *Vibrio* sp. and choleric toxin (*ctx*) *Vibrio cholerae*, associated with the sources of contamination in the Mandinga Coastal Lagoon (MCL). During 2017, samplings were carried out in which sources of contamination associated with anthropogenic activities were identified. At the same time, water samples were collected from which DNA was extracted and the presence/absence of the HA and *ctx* genes was detected with a PCR analysis. The HA gene was identified in the three seasons of the year, while the *ctx* gene was only present in the dry and rainy seasons. The prevalence of both genes in the study area was independent of the presence of the pollution sources identified in the area. The absence of the *ctx* gene during the northern season is associated with the variability of the physicochemical parameters typical of the season.

## 1. Introduction

The state of Veracruz has 16 coastal lagoons of ecological and economic significance along its coastline [1]. These water bodies exhibit estuarine characteristics, with salinity varying from mesohaline to polyhaline, influenced by hydrological features such as seasonality, tidal exchange, and freshwater influx [2]. Veracruz’s coastal lagoons are essential to produce commercially valuable marine and freshwater organisms, whether in open or closed systems, and for aquaculture conducted in cages or ponds [3]. Among these, the Mandinga Coastal Lagoon (MCL) stands out as one of the most important producers of fishery resources such as oysters, crab, and finfish fishery along the Gulf of Mexico coast [4].

In addition to their productivity, coastal lagoons provide essential environmental services [5,6]. However, these services have been adversely affected by factors such as urbanization and pollution due to cultural and socioeconomic changes. These shifts have led to the development of residential complexes along the lagoon’s periphery, contributing to waste discharge and to increased tourism and commercial activities, which require both domestic and industrial water consumption, subsequently resulting in wastewater production [1,4].

The lack of wastewater treatment can be a significant source of contamination in countries like Mexico, where average annual temperatures and salinity in warm-humid climate regions range between 27.67 and 31.38 °C and 6.27 and 24.18‰ presenting a risk of waterborne disease transmission [7]. Discharges from residential developments, insufficient sanitation measures, and agricultural contaminants have resulted in point sources (urban and industrial waste) and diffuse sources (agrochemical waste). Consequently, it is not surprising to find enterobacteria in filter organisms like oysters and unpleasant odors in the lagoon [8]. Furthermore, surface waters are exposed to meteorological or climatic variations, with climate change likely to raise water temperatures and increase flooding events, thereby elevating the risk of disease transmission to vulnerable coastal populations. This is particularly concerning for bacteria that adapt to aquatic environments, such as *Aeromonas* spp. and *Vibrio* spp. [9].

*Vibrio cholerae* is a pathogenic enterobacterium of the *Vibrio* genus known to cause a gastroenteric syndrome. It promotes hypersecretion of ions and water in the small intestine and can be highly harmful to humans, especially those who come into direct contact with or consume raw seafood from tropical estuarine environments with significant salinity variations [10,11]. *V. cholerae* is also associated with cholera epidemics [12]. The pathogenicity of this serogroup is attributed to the presence of the cholera toxin gene (*ctx*); however, not all bacteria of this genus carry the *ctx* virulence genes, though they may contain other toxins that cause gastrointestinal inflammation upon ingestion. Therefore, the presence of these bacteria in bodies of water with salinity variability could serve as a reservoir for the spread of virulent strains, posing a potential epidemic risk [10]. One way to detect the cholera toxin gene in *Vibrio cholerae* is through molecular characterization or analysis of *V. cholerae* strains collected from aquatic environments [13,14].

In 2009, the National Commission for Knowledge and Use of Biodiversity (CONABIO) conducted a study to identify mangrove sites of biological relevance and those in need of rehabilitation [15]. This study provided insights into the growing anthropogenic development of the area. Therefore, the aim of this study was to identify the presence of the *Vibrio* sp.-specific zinc metalloproteinase (HA) gene and the *Vibrio cholerae*-specific *ctx* gene associated with pollution sources in the Mandinga Coastal Lagoon.

## 2. Materials and Methods

### 2.1. Study Area

This research was conducted in the Mandinga Coastal Lagoon (MCL), located in the municipality of Alvarado, Veracruz, Mexico, between latitudes 19°00′ and 19°06′ North and longitudes 96°02′ and 96°06′ West. The lagoon system runs north–south, while the nearby coastline is oriented northwest-southeast, forming the tip of Antón Lizardo and borders Boca del Río municipality to the north and Medellín de Bravo to the southwest (Figure 1). The system covers an area of 3250 hectares and spans a total length of 20 km [16]. It is connected to the Atoyac and Jamapa river basins, which link the Sierra Madre Oriental with the coastal plains of the Gulf of Mexico.

The prevailing climate in the area is warm sub-humid or semi-humid with summer rains, with an average maximum temperature of 28.9 °C and a minimum of 17.2 °C, and an annual precipitation range of 1731 mm [17]. The MCL experiences a dry season (March to June), a rainy summer season (July to October), and a windy season (November to February) [18,19].

### 2.2. Identification of Pollution Sources

To identify sources of pollution, a characterization of the MCL was performed by comparing satellite images from 2007 and 2017 to locate areas of highest population growth (Figure 2). Using this information, a reconnaissance visit was conducted to refine the sampling design and to identify potential pollution sources and activities in the area. Information was gathered on local activities, the area’s history, and issues perceived by local fishers, with the assistance of the “Cooperativa Fraternidad”. This collaboration helped identify both point and diffuse sources discharging into the MCL. Key informants provided data on sites of interest, which were cross-referenced with bibliographic information about the area. Based on these findings, sampling sites were selected to best represent the different activities conducted in the area.

### 2.3. Selection and Location of Sampling Sites

Based on the reconnaissance visit and collected information, the MCL was divided into three zones for this study, with 10 sampling sites selected based on their influence from pollution sources and their distribution throughout the system. Zone 1 (one site) encompasses Grande Lagoon; Zone 2 (five sites) covers Redonda Lagoon; and Zone 3 (four sites) includes Larga Lagoon. All sites were georeferenced (Table 1).

### 2.4. Sample Collection

Sampling was carried out during the rainy season, north winds, and dry season, with six water samples collected at each site following the guidelines of current regulations, which specify sampling at a depth of 15 to 30 cm. Samples were collected in sterile bags (Whirlpak) of 120 mL were used. All water samples were refrigerated at 4° C [20]. In situ measurements of physicochemical parameters, including water temperature, pH, and salinity, were recorded using a YSI 6600 multiparameter probe (584 Park East Drive Woonsocket, RI 02895, USA). Samples were refrigerated at ±4 °C and transported to the Research and Postgraduate Laboratory (LIRA) at the Technological Institute of Boca del Río for processing within 4 h of collection.

### 2.5. Molecular Identification

After water samples were collected, they were pre-enriched in flasks with 90 mL of alkaline peptone water (APA) + 3% NaCl, adjusted to a pH of 8.4. Each flask was inoculated with 10 mL of sample and incubated for 24 h at 35 °C. After incubation, the flasks were vortexed to homogenize the contents, and 1 mL was extracted for bacterial DNA using the Wizard^®^ Genomic DNA Purification kit (Promega Corporation; Madison, WI, USA). Optimal conditions for PCR amplification were determined through preliminary assays modifying the FDA Bacteriological Analytical Manual: *Vibrio* specifications, and to the specifications of other authors [21,22], resulting in the following concentrations (Table 2).

PCR analysis was conducted to identify the zinc metalloproteinase gene of the Vibrionaceae family [23,24]. For positive samples, the presence of the cholera toxin gene (*ctx*) was further identified.

The oligonucleotides used for the polymerase reaction are shown in Table 3, including the gene for zinc metalloproteinase (HA) specific to the Vibrionaceae family and the *ctx* gene specific to *Vibrio cholerae*.

PCR assays were performed using PROMEGA reagents, with amplification reactions conducted on an Apollo ATC201 thermocycler. Conditions for *Vibrio* sp. reactions are shown in Table 4, and those for *V. cholerae* in Table 5.

For each PCR amplification product, 7 μL were loaded onto a 1% agarose gel (Promega) containing ethidium bromide, a DNA stain, to enable visualization. The gel was then placed in an electrophoresis chamber with 1X TAE buffer (composed of Tris, acetic acid, and EDTA) and subjected to a constant voltage of 90 V for 60 min. To estimate fragment sizes, a 1000 bp molecular weight marker was used alongside positive and negative controls. Finally, the DNA bands were visualized under ultraviolet light using a UVP transilluminator. Successful amplifications were indicated by DNA fragments of approximately 225 bp for *Vibrio* sp. and 777 bp for *V. cholerae*.

### 2.6. Statistical Analysis

Descriptive statistics were used to analyze the physicochemical parameters recorded in the Mandinga coastal lagoon, located in Alvarado, Veracruz, Mexico. In addition, a non-parametric comparative analysis Mann–Whitney U test (*p* ≤ 0.05) was applied to identify significant differences between the sources of contamination and the presence of the HA gene, specific to *Vibrio* sp., and the *ctx* gene, characteristic of *Vibrio cholerae*, recorded at the sampling sites during the study period. The analyses were performed using the Jamovi program^®^ (version 2.3.26).

## 3. Results

### 3.1. Identification and Description of Point Sources in the Study Area

In Zone 1 (S1), water is primarily used for fishing and some aquaculture activities, while the land serves partly as an environmental protection area, preserving mangrove and wetland zones. However, nearby housing developments like the Lomas de la Rioja II subdivision have led to deforestation around the lagoon due to heavy machinery use, causing environmental degradation (Table 6). Site S1 was considered the least impacted, with diffuse pollution sources identified in the area (Table 7).

Zone 2 (S2, S3, S4, S5, and S6) experiences higher levels of human activity, especially around the locality of Mandinga. Water use is mainly divided between two types of activities: (1) tourism, including boat tours, and (2) primary production activities such as oyster farming and shrimp cultivation. The land in this area is predominantly used for residential and commercial purposes, with many local restaurants and cooperative groups dedicated to fishing and the extraction of aquatic resources, especially oysters. An oyster purification plant is also located here. Some mangrove areas remain intact, while others have been converted to agricultural use, primarily for pineapple cultivation (Table 6). Point-source pollution was identified from wastewater outlets near restaurant areas that discharge directly into the lagoon, alongside diffuse sources, such as the pineapple cultivation area (Table 7).

In Zone 3 (S7, S8, S9, and S10), the primary land use consists of residential subdivisions and a commercial plaza, El Dorado. Water activities are mainly tourism-oriented, such as boat tours and recreational watercraft like jet skis. Many residences have private docks and visible drainage outlets. The land use here is primarily urban development, supporting both commercial and residential needs. However, near the connection between Redonda Lagoon and Larga Lagoon, an area is used for cattle grazing (Table 6). In this zone, point-source pollution sources were identified, including direct discharges from homes and the shopping center, along with diffuse sources, such as the grazing area (Table 7).

### 3.2. Physicochemical Parameters of Water

During the dry season, the highest average water temperature was recorded at Site 7. The lowest temperature was observed at Site 10. In the rainy season, the highest recorded temperature was at Sites 4 and 5, while the lowest was at Site 9. During the windy season, the maximum average water temperature was measured at Sites 7 and 8, with the other sites recording a consistent 25 °C. In terms of pH, the highest average value during the dry season was at Site 8, while the lowest was at Site 1. During the rainy season, Site 5 showed the highest pH, and Site 1 had the lowest. In the windy season, the maximum pH was recorded at Site 6, while the lowest was at Site 1. Salinity peaked during the dry season at Site 9, while the lowest was at Site 1. In the rainy season, the highest average salinity was recorded at Site 8, and the lowest was at Site 10. During the windy season, the highest salinity was measured at Site 10, and the lowest at Site 4 (Table 8).

### 3.3. HA Genes of Vibrio sp. and of the ctx Gene of Vibrio cholerae

Positive results for the *Vibrio* sp.-specific zinc metalloproteinase (HA) gene were obtained at sites S4, S5, S6, S7 and S8 during the dry and rainy seasons. Likewise, sites S6, S7 and S8 were positive for *Vibrio* sp. during the northern season. At site S5, the *Vibrio* sp.-specific zinc metalloproteinase (HA) gene and the *V. cholerae*-specific cholera toxin (*ctx*) gene were detected (Table 9).

The PCR analysis results are shown in Figure 3. Lane M displays the 1000 bp marker ladder, Lane C-includes a negative control, Lanes 1 to 6 contain sample results, and Lane C+ has a positive control for *Vibrio* sp. In Lane 6, a 225 bp band was amplified, corresponding to *Vibrio* sp. (Figure 3a). In Figure 3b, Lane C+ shows the positive control for the cholera toxin gene (*ctx*). Lane 1 shows a 777 bp band, which corresponds to *V. cholerae* (Figure 3b).

Regarding the relationship between *Vibrio* sp. and *V. cholerae* and both point and diffuse contamination sources in the Mandinga Coastal Lagoon (MCL), Zone 1 yielded negative results for both bacteria. This zone is used primarily for fishing activities and aquaculture enclosures, with some protected environmental areas remaining intact. In Zone 2, where point sources included wastewater discharges from restaurants and residences and diffuse sources stemmed from agricultural activities like pineapple cultivation, both bacteria were detected. Specifically, Site 5 showed the presence of both bacteria during the dry and rainy seasons, though results were negative during the windy season. At Site 6, *Vibrio* sp. was present during the dry and windy seasons, but *V. cholerae* was not detected.

Zone 3, characterized by extensive residential, tourism, and commercial activities, also showed positive results for *Vibrio* sp. Site 7 was positive during the dry and windy seasons, while Site 8 showed positivity during the rainy and dry seasons. However, no *V. cholerae* was detected in this zone. The primary point sources of contamination here were direct discharges from residences and the commercial center, with diffuse sources linked to livestock grazing areas.

The results showed a higher prevalence of the presence of the HA gene specific to *Vibrio* sp. in the disuse, diffuse and point sources. However, the statistical analysis carried out determined that there are no significant differences (*p* > 0.05) between the sources of contamination and the detection of the presence of the HA gene *p* = (0.270) and the *ctx* gene specific to *V. Cholerae p* = (0.307) during the study period.

## 4. Discussion

The zoning conducted in this study was based on land and water usage patterns, which reflect the various anthropogenic activities occurring in the Mandinga Coastal Lagoon (MCL). Coastal lagoons are productive, biodiverse environments with high-quality landscapes, supporting diverse activities [25,26]. This is consistent with the land use trend reported in the area by another study, such as anthropogenic and agricultural-livestock development, where anthropogenic development continues to be a growing constant [27]. Unfortunately, these activities are associated with the production of sources of pollution, as revealed by a study in the coastal area adjacent to the MCL, where 88 storm drains were identified, 50 drains with the presence of wastewater discharge and four drains with the presence of unidentified water. The main economic activities identified were tourism, commerce, fishing and services [28]. During the dry season, a positive correlation was found between the presence of hotels and the presence of point source pollution discharges on the surrounding beaches of the Veracruz-Boca del Río coastal zone; these pollution sources have a dispersion radius of 31.398 km to 73.965 km of influence around the zone. Therefore, it was determined that the persistence of nearby wastewater can affect the transparency and dissolved oxygen of the water, facilitating the growth of pathogenic organisms in the area [29].

In the vicinity of Zone 1 (Grande Lagoon), diffuse pollution sources were identified, with signs of land subdivision and the presence of heavy machinery used for street construction, lighting, and sewer systems. However, in the Grande Lagoon sampling site, land and water use is primarily designated for fishing and environmental protection, making it the least impacted zone. It was thus used as a reference site for other samplings since areas of natural vegetation, particularly mangrove forests, are still conserved here. This low-impact status likely contributed to the absence of *Vibrio* sp. and *V. cholerae*.

The MCL has natural and coastal areas with characteristics of economic interest and a privileged location, where urban growth is a constant. Therefore, it is recommended to periodically carry out evaluations on the change in land use, which allows the protection and preservation of this ecosystem [27]. The implementing strategies aimed at protecting coastal wetlands in buffer zones near human settlements could reduce pollutant discharges, particularly during climatic events, and that measures raising awareness of environmental protection activities could help decrease water contamination in coastal systems [30].

Other authors argue that coastal lagoons are typically impacted by human activities like wastewater discharges, hydrological modifications, and land use changes [31], which can disrupt ecological functioning, system self-regulation, and provision of ecosystem services [5]. There are antecedents of the presence of Vibrionaceae in the coastal lagoon of Mandinga [32,33]. In this study, the HA gene specific to *Vibrio* sp. was identified during the three seasons of the year and the *ctx* gene during the dry and rainy seasons in zones 2 and 3. The presence of the specific gene for *V. Cholerae* may vary with seasonal changes in environmental parameters [33]. In addition, these zones presented point and diffuse sources of contamination. This indicates that this type of bacteria is typical of these ecosystems that have a tendency to change land use, which generates a negative impact that favors the presence of genes related to the HA and *ctx*. These bacteria in these environments in other parts of the world are considered ubiquitous organisms [34]. This shows that brackish water bodies are reservoirs of this type of bacteria [26,35]. These results align with findings from other authors, who observed human-pathogenic *Vibrio* species present year-round, indicating a continuous potential for infection. They also detected three potentially pathogenic *Vibrio* strains in coastal waters [36].

The increase in organic matter during summer in coastal lagoons stems from human activities that produce dissolved and suspended particulate matter, originating from agricultural and industrial runoff [26]. These conditions could correlate with the presence of *Vibrio* in Zone 2 of the MCL, which had average temperatures of 20–29 °C, pH up to 7.9, and salinity levels between 18 and 20 PSU. In Site 5, where the cholera toxin gene was detected in the dry and rainy seasons, higher organic matter levels may result from shrimp and oyster farming, which serve as reservoirs for *Vibrio* strains, posing potential risks to consumers, as described by other authors [22,37]. They identified *Vibrio* spp. in *Crassostrea virginica* oyster samples from other Veracruz lagoon systems, with *V. cholerae* strains detected during the north wind season in oysters obtained from cooperatives. This suggests that *Vibrio* strains show adaptability to low temperatures and the ability to colonize organisms post-harvest for consumption.

That aquaculture activities in coastal ecosystems maintain high salinity, suspended solids, organic matter, and bacteria similar to *Vibrio* spp. [38,39]. The presence of *Vibrio* spp. in cultivated organisms poses infection risks when consumed raw, potentially causing gastrointestinal illness [40]. Socioeconomic activities such as tourism could also have an impact, given that Mandinga is the most densely populated area in the study [1,41].

*Vibrio* presence is also linked to lagoon depth; in another study, significant differences were determined between water samples taken at the surface and at the bottom, with sediment resuspension promoting the presence of these bacteria in the water column [42]. The MCL has an average depth of 1 m, with maximum depths of 3.6 m and minimum levels of 0.5 m, categorizing it as a shallow system [43]. The shallow depth and water currents could have influenced *Vibrio* sp. presence in the sampling sites. Other researchers found higher concentrations of *Vibrio* spp. and *Enterococcus* spp. in sediments compared to the water column [44]. They also found that bacteria like *E. coli*, *Salmonella paratyphi*, and *V. parahaemolyticus* survive better in sediment than in surface water, likely due to sediment characteristics that enhance pathogen survival. This is explained by the fact that in the sediment, the particle size as well as the high content of organic carbon enhance the survival of indicator and pathogenic bacteria [45]. That the water column is composed of up to 71% organic matter, while sediments are mainly composed of protein-like components and provide protection and nutrient supply and adhesion substrates for these types of bacteria [26,44].

During the study period in the MCL, activities like mechanical dredging, boat tours, anchoring with wooden posts, and manual shellfish collection disturbed the bottom, promoting sediment resuspension and water quality degradation. Besides depth, *Vibrio* presence in Sites 4, 5, 6, 7, and 8 may have been influenced by wastewater discharges from tourist and residential areas, which, while lacking detailed composition data, may provide nutrients that promote microbial growth [46]. Zone 3 had two positive sites, S7 and S8, showing *Vibrio* sp. presence but no *V. cholerae*.

Regarding anthropogenic activity, Zone 3 has fewer inhabitants since many residential subdivisions are used as vacation homes, resulting in a lower population density than Zone 2. This zone, located near the sea, also contains the navigation channel and is the deepest area in the system. Spatial-temporal variations in *Vibrio* sp. patterns in our study may relate to suspended particle concentrations, indicating the importance of determining an appropriate sampling depth, as recommendations often favor sediment-based sampling [47]. It is important to highlight that *Vibrio* species in coastal marine systems form a bacterial group involved in various interactions with other ecosystem components, contributing significantly to the life cycles of different marine organisms. However, some *Vibrio* species are pathogenic [48]. Even though environmental isolates of *V. cholerae* may occasionally lack the *ctx* gene, this does not rule out the potential for its presence [49].

The detecting *Vibrio* strains and *V. cholerae*-related genes via PCR techniques in coastal lagoons can help manage microbial risks from contaminated shellfish [50]. Continued environmental education through scientific research is crucial to fostering environmental discipline and sustainable coastal resource management [51]. Regulatory authorities, marine resource producers, and health professionals must also monitor coastal lagoons to mitigate bacterial risks [52]. In the MCL, proper hygiene practices, responsible fishery management [53], and environmental monitoring are essential for maintaining resource health and public safety [8].

## 5. Conclusions

Identifying the sources of both point and diffuse contamination in the Mandinga Coastal Lagoon allowed us to describe shifts in land and water use, helping to pinpoint areas with the highest anthropogenic activity. The prevalence of the zinc metalloproteinase (HA) gene specific for *Vibrio* sp., and the cholera toxin gene (*ctx*) specific for *Vibrio cholerae* are independent of the presence of the diffuse and point sources of contamination identified in the area. However, factors such as negative environmental impact can increase the tendency of the presence of the genes of these bacteria in highly impacted environments. The presence of the HA gene *Vibrio* sp. is independent of the season of the year since positive results were obtained in the three seasons sampled, while the absence of the *ctx* gene *Vibrio cholerae* in the northern season is associated with the variability of the physicochemical parameters typical of the season. The spatial distribution of both bacteria is related to activities such as oyster farming, shrimp farming, pineapple farming, discharge of wastewater from restaurants, direct discharge of wastewater from residential/commercial homes, and cattle grazing areas.

## Figures and Tables

**Figure 1 microorganisms-13-00352-f001:**
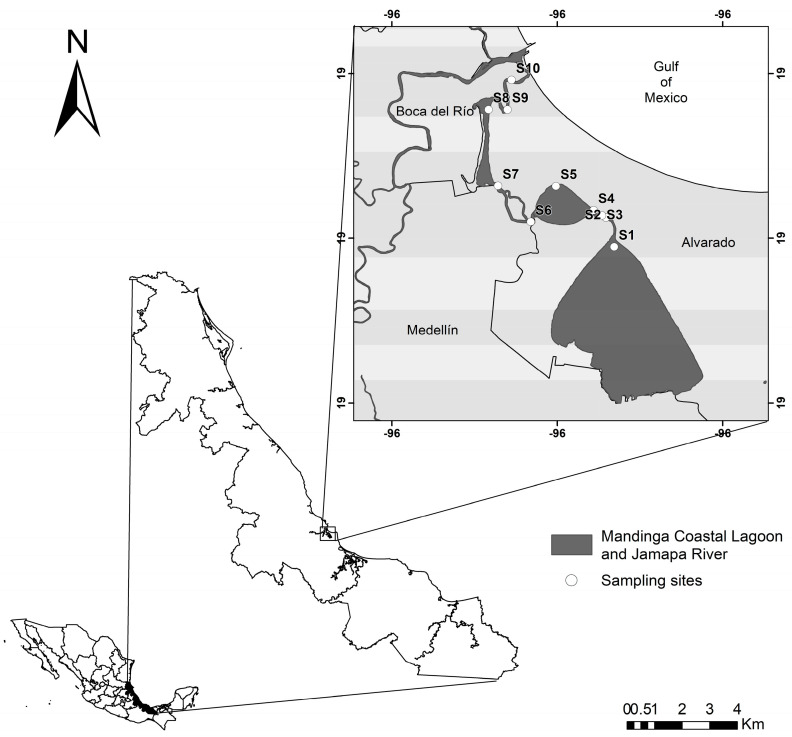
Study area and sampling site locations in the Mandinga Coastal Lagoon, Alvarado, Veracruz, Mexico.

**Figure 2 microorganisms-13-00352-f002:**
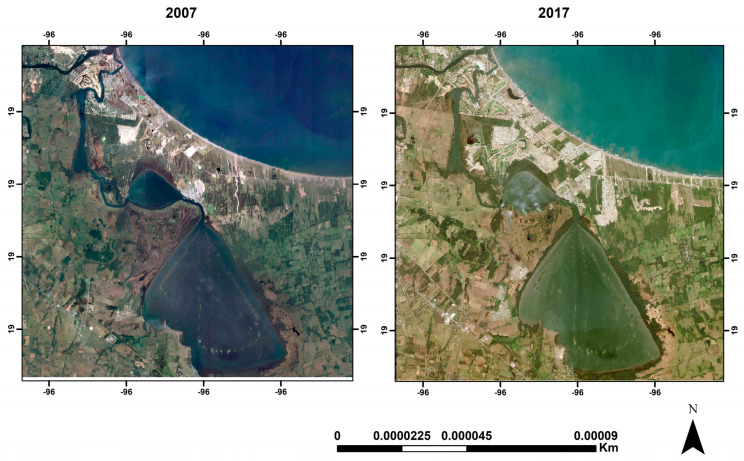
Comparison of satellite images of the Mandinga Coastal Lagoon, Alvarado, Veracruz, Mexico.

**Figure 3 microorganisms-13-00352-f003:**
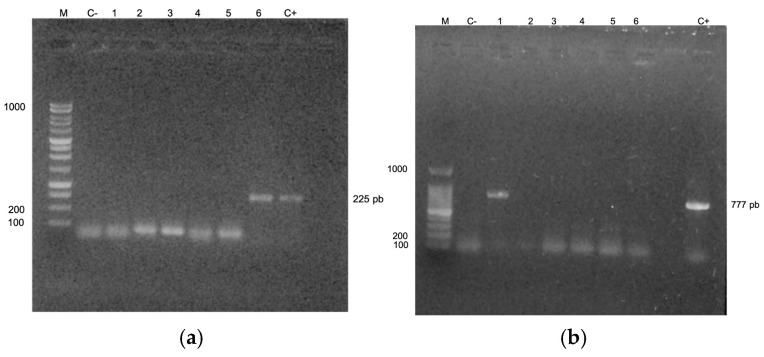
Presents the PCR results: (**a**) shows the identification of the genes for zinc metalloproteinase (HA) *Vibrio* sp., while (**b**) confirms the presence of the choleric toxin (*ctx*) in *V*. *cholerae*. In the Mandinga Coastal Lagoon water samples.

**Table 1 microorganisms-13-00352-t001:** Geographic coordinates of selected sampling sites.

Site	**Latitude**	**Longitude**	**Description**
1	19.0474	−96.0736	Mangrove area
2	19.0378	−96.0709	Aquaculture area
3	19.0481	−96.0749	Oyster purification plant
4	19.0498	−96.0776	Restaurant area
5	19.0577	−96.0899	Mangrove area
6	19.0460	−96.0981	Boca de Matosa
7	19.0578	−96.1088	Punta Tiburón residential area
8	19.0828	−96.1119	Villa Rica Golf Club
9	19.0828	−96.1057	El Conchal residential area
10	19.0925	−96.1044	El Dorado shopping center

**Table 2 microorganisms-13-00352-t002:** Reagent concentrations used for PCR.

Reagent	Concentration
Buffer	1X
MgCl2	1.5 mM
dNTPs	200 mM
Forward Primer (F)	0.5 μM
Reverse Primer (R)	0.5 μM
DNA	40 ng
GoTaq	2.5 U

**Table 3 microorganisms-13-00352-t003:** Oligonucleotide sequences used in PCR reactions conditions for the detection of HA and *ctx* gene.

Primer	Sequence 5′-3′	bp	Reference
HA-F	CATGAGGTCAGCCACGGTTTTACTGAGCAG	225	Sussman, M., Willis, B.L, et al., 2008 [23]
HA-R	CGCGCGGTTAAACACGCCACTCGAATGGTGAAC		Sussman, M., Willis, B.L, et al., 2008 [23]
CTa	TGA AAT AAA GCA GTC AGG TG	777	FDA-BAM [24]
CTb	GGT ATT CTG CAC ACA AAT CAG		FDA-BAM [24]

**Table 4 microorganisms-13-00352-t004:** Amplification conditions for the detection of HA gene of *Vibrio* sp.

Phase	No. of Cycles	Time	Temperature
Denaturation	1	5 min	94 °C
		20 s	94 °C
Amplification	30	20 s	55 °C
		1 min	72 °C
Extension	1	5 min	72 °C

**Table 5 microorganisms-13-00352-t005:** Amplification conditions for the detection of the *ctx* gene of *Vibrio cholerae*.

Phase	No. of Cycles	Time	Temperature
Denaturation	1	3 min	94 °C
		1 min	94 °C
Amplification	35	1 min	55 °C
		1 min	72 °C
Extension	1	3 min	72 °C

**Table 6 microorganisms-13-00352-t006:** Identification of anthropogenic activities in the Mandinga Coastal Lagoon, Alvarado, Veracruz.

Zone	Water Use	Land Use
1	Rural fishing activities	Environmental protection
	Aquaculture enclosures	Residential
2	Tourism activities	Housing
	Oyster and	Commercial
	Shrimp farming	Agricultural
3	Residential	Residential
	Tourism and	Commercial
	Recreational use	Livestock grazing

**Table 7 microorganisms-13-00352-t007:** Pollution sources in the Mandinga Coastal Lagoon, Alvarado, Veracruz.

	Lagoon	Site	Description	Pollution Source
1	Grande	1	Mangrove area	Diffuse
2	Redonda	2	Aquaculture area	Point and Diffuse
		3	Oyster purification plant	
		4	Restaurant area	
		5	Mangrove area	
		6	Boca de Matosa	
3	Larga	7	Punta Tiburón Residential area	Point and Diffuse
		8	Villa Rica Golf Club	
		9	El Conchal Residential area	
		10	El Dorado Shopping Center	

**Table 8 microorganisms-13-00352-t008:** Monthly means for the water physicochemical parameters in the Mandinga Coastal Lagoon, Alvarado, Veracruz.

	T (°C)	pH	Salinity (UPS)
Dry season			
Mean	28.13 ± 0.35	7.93 ± 0.205	25.03 ± 1.13
Maximum	28.45 ± 0.21	8.15 ± 0.07	26.70 ± 0.007
Minimum	27.35 ± 0.07	7.45 ± 0.07	23.46 ± 0.014
Rainy season			
Mean	28.23 ± 0.75	7.64 ± 0.22	15.99 ± 1.59
Maximum	29.05 ± 0.07	7.950 ± 0.07	18.54 ± 0.148
Minimum	26.950 ± 0.07	7.25 ± 0.07	12.80 ± 0.056
Windy season			
Mean	25.20 ± 0.406	7.96 ± 0.051	25.35 ± 0.92
Maximum	26 ± 0.00	8.02 ± 0.005	22.32 ± 0.01
Minimum	25 ± 0.00	7.84 ± 0.036	20.05 ± 0.05

T °C: Temperature; pH: Hydrogen potential; PSU: Practical Salinity Unit.

**Table 9 microorganisms-13-00352-t009:** Annual results showing presence (+) or absence (−) of HA gene of *Vibrio* sp. and of the *ctx* gene of *Vibrio cholerae* in sampling sites by zone.

Zone	Site	Dry Season		Rainy Season		Windy Season	
		*Vibrio* sp.	*V. cholerae*	*Vibrio* sp.	*V. cholerae*	*Vibrio* sp.	*V. cholerae*
1	1	−	−	−	−	−	−
2	2	−	−	−	−	−	−
	3	−	-	−	−	−	−
	4	+	−	+	−	−	−
	5	+	+	+	+	−	−
	6	+	−	−	−	+	−
3	7	+	-	−	−	+	−
	8	−	−	+	−	+	−
	9	−	−	−	−	−	−
	10	−	−	−	−	−	−

## Data Availability

The original contributions presented in the study are included in the article, further inquiries can be directed to the corresponding author.

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
