# Peer review of "Vibrio sp. and Identification of the ctx Gene of Cholera Toxin in the Mandinga Coastal Lagoon, Veracruz, Mexico"

_microorganisms, 2025, doi:10.3390/microorganisms13020352_

Round 1

Reviewer 1 Report

Comments and Suggestions for Authors

Тhe article about one of the most important areas of the Gulf of Mexico where many of the region's seafood products, such as oysters, crabs and fish, are produced. The study of the qualitative state of the aquatic environment of this region is relevant. The purpose of this study is to assess the presence of Vibrio spp. and identify the gene responsible for cholera toxin in V. cholerae in relation to human activities in the lagoon. The results show that Vibrio spp. is present in the lagoon water all year. A clear link was discovered between the presence or absence of Vibrio spp./V. cholerae and human activities. The authors by this study propose a method for assessing the presence of Vibrio spp. and V. cholerae, which is important for monitoring the quality of the environment.

The article is interesting, I support this work, but I recommend checking the English style of presentation with native speakers, as well as reading the rules for authors and improving the manuscript, starting with the abstract, then the methodology and discussion.

The abstract should be a total of about 200 words maximum. The abstract should be a single paragraph and should follow the style of structured abstracts, but without headings: 1) Background: Place the question addressed in a broad context and highlight the purpose of the study; 2) Methods: Describe briefly the main methods or treatments applied. Include any relevant preregistration numbers, and species and strains of any animals used; 3) Results: Summarize the article's main findings; and 4) Conclusion: Indicate the main conclusions or interpretations. The abstract should be an objective representation of the article: it must not contain results which are not presented and substantiated in the main text and should not exaggerate the main conclusions.

Introduction. The goal needs to clarify and indicate with which of the types of activities the authors hypothesized to associate the emergence of vibrio toxins

Methods. I did not find a description of statistical processing of the material in the methods. Also, explain what "spp" means - is it several species? It might be better to write the generic name everywhere, if it is not essential.

Results. Still, it is necessary to somehow statistically link diffuse pollution sources and the detection of vibrio toxin genes. Correlation analysis.

Amaral et al. (2023) [23] should be Amaral et al.  [23]. Check references everythere in text.

In the conclusions, it is still necessary to clarify with which of the types of activities the authors associate the appearance of vibrio toxins like mechanical dredging, boat tours, anchoring with wooden posts, and manual shellfish collection disturbed the bottom, promoting sediment resuspension and water quality degradation.

Author Response

Respuesta al revisor 1 Comentarios

Estimado revisor 1:

Muchas gracias por tomarse el tiempo de revisar este manuscrito. Sus observaciones y recomendaciones nos han ayudado a refinar nuestra investigación y mejorar la calidad general del manuscrito. A continuación, encontrará las respuestas detalladas y las correcciones correspondientes resaltadas en color rojo. Consulte el archivo adjunto.

Comentarios 1:  El artículo es interesante, apoyo este trabajo, pero recomiendo revisar el estilo de presentación en inglés con hablantes nativos, así como leer las normas para autores y mejorar el manuscrito, empezando por el resumen, luego la metodología y la discusión. Respuesta 1: Estamos de acuerdo con este comentario. Agradecemos su apoyo al artículo. Por ello, se revisó a profundidad la guía de autores de la revista para adaptar el documento a las pautas de cada sección, principalmente, el resumen, materiales y métodos y la discusión como se solicitó. El manuscrito ha sido revisado en cuanto al estilo de presentación y fue mejorado por un traductor con experiencia en el idioma.

Comentarios 2: El resumen debe tener una extensión total de 200 palabras como máximo. El resumen debe ser un solo párrafo y debe seguir el estilo de los resúmenes estructurados, pero sin encabezados: 1) Antecedentes: Sitúe la pregunta abordada en un contexto amplio y resalte el propósito del estudio; 2) Métodos: Describa brevemente los principales métodos o tratamientos aplicados. Incluya cualquier número de prerregistro relevante y especies y cepas de cualquier animal utilizado; 3) Resultados: Resuma los hallazgos principales del artículo; y 4) Conclusión: Indique las principales conclusiones o interpretaciones. El resumen debe ser una representación objetiva del artículo: no debe contener resultados que no se presenten y fundamenten en el texto principal y no debe exagerar las conclusiones principales. Respuesta 2: Gracias por señalar esto. Estamos de acuerdo, hemos modificado en consecuencia. Se realizó una revisión exhaustiva del manuscrito y, de acuerdo con las pautas indicadas, se reestructuró y modificó el resumen para adaptarlo al formato. Este cambio se puede encontrar en la sección “Resumen” página 1, líneas 10 a 23.

Comentarios 3: Introducción. El objetivo debe aclarar e indicar con qué tipo de actividades los autores plantearon la hipótesis de que se asociaba la aparición de toxinas de vibrio. Respuesta 3: Gracias por señalarlo. Estamos de acuerdo, lo hemos modificado en consecuencia. El objetivo fue reestructurado y modificado de acuerdo con la hipótesis planteada para esta investigación. Este cambio se puede encontrar en la sección “Introducción”, página 2, párrafo 5, líneas 70 a 73.

Comentarios 4: Métodos. No encontré en los métodos una descripción del procesamiento estadístico del material. Además, explique qué significa “spp” - ¿se trata de varias especies? Tal vez sea mejor escribir el nombre genérico en todas partes, si no es imprescindible. Respuesta 4: Gracias por señalarlo. Estamos de acuerdo con este comentario. Por lo tanto, se realizó un análisis para seleccionar una prueba estadística adecuada. Considerando el diseño de nuestra investigación, mediciones, pregunta e hipótesis, se seleccionó una prueba no paramétrica U de Mann-Whitney (p ≤ 0,05) para realizar un análisis comparativo de la media y determinar si existen diferencias considerando como variable dependiente la presencia del gen HA, específico de Vibrio sp., y el gen CTX, característico de Vibrio Cholerae , y como factores las fuentes de contaminación. Este cambio se puede encontrar en la sección “Materiales y métodos” 2.6. Análisis estadístico, página 5, párrafo 1, línea 167 a 175. Con respecto a explicar la palabra “spp” o cambiarla por “sp”, se analizó el concepto y consideramos que lo más adecuado en esta investigación para referirse a las bacterias del género Vibrio debe ser con “sp”. Agradecemos su aporte. Por ello, en este manuscrito se sustituyó la palabra “spp.” por “sp”.

Comentarios 5: Resultados. Aún así, es necesario relacionar de alguna manera estadísticamente las fuentes difusas de contaminación y la detección de genes de toxinas de vibrio. Análisis de correlación. Respuesta 5: Considerando nuestro diseño de investigación, pregunta e hipótesis, se seleccionó una prueba no paramétrica U de Mann-Whitney (p ≤ 0.05) para realizar un análisis comparativo de la media y determinar si existen diferencias considerando como variable dependiente la presencia del gen HA, específico de Vibrio sp. y el gen CTX, característico de Vibrio Cholerae , y como factores las fuentes de contaminación. Los resultados estuvieron de acuerdo con otros estudios, ya que se demostró que los genes HA y CTX son independientes tanto de las fuentes difusas como de las puntuales de contaminación. Esto indica que este tipo de bacterias son propias de este tipo de ecosistemas. Este cambio se puede encontrar en la sección “Resultados” 3.3. Identificación molecular, página 9, párrafo 5, línea 266 a 270.

Comentarios 6: Amaral et al. (2023) [23] debería ser Amaral et al. [23]. Verificar las referencias en todo el texto. Respuesta 6: Estamos de acuerdo con este comentario. La sección “Discusión” fue revisada y modificada de acuerdo con las pautas de la guía de autores para la citación de referencias. Este cambio se puede encontrar en la sección “4. Discusión”, página 9, línea 272 a 380. Además, esta sección fue modificada: se agregaron artículos de otros investigadores a la discusión. Este cambio se puede encontrar en la página 9, párrafo 1, línea 273 a 389; párrafo 3, 297 a 300; página 9-10, párrafo 4, línea 308 a 320.

Comentarios 7: En las conclusiones, aún es necesario aclarar con cuáles de los tipos de actividades los autores asocian la aparición de toxinas de vibrio como el dragado mecánico, los paseos en lancha, el fondeo con postes de madera y la recolección manual de mariscos, perturbaron el fondo, promoviendo la resuspensión de sedimentos y la degradación de la calidad del agua. Respuesta 7: Gracias por señalar esto. Estamos de acuerdo con este comentario. La información de la sección de conclusiones fue reestructurada y modificada de acuerdo con los lineamientos del formato y los resultados de la investigación. Además, se consideró su recomendación, por lo que se agregaron las actividades antropogénicas asociadas con la prevalencia del gen específico de Vibrio sp. (HA) y el gen específico de la toxina de V. cholerae (CTX). Este cambio se puede encontrar en la sección “5. Conclusiones” página 11, líneas 378 a 392.

Reviewer 2 Report

Comments and Suggestions for Authors

The authors report the presence of Vibrio Spp. in the Mandinga Lagoon System, Veracruz, Mexico. The reviewer suggests the rejection of this paper due to the following reasons.

1) The quantity of the results is not enough for the hull article. Only the presence of two genes was presented without quality control.

2) The quality of the language is below the standard of international journals. There are many typing errors.

3) The scientific presentation skill by the authors is low.

3-1) Section “3.1 Water and Land Use” can be moved to the materials and methods section, because section 3.1 is a part of the description of the target watershed.

3-2) In the section of “3.2 Physicochemical Parameters of Water”, the authors can prepare a figure or a table to present their measurements.

3-3) Past tense must be used for the citation in the text. For example, “Amaral et al. (2023) [23] state that the increase in organic matter …” > “Amaral et al. (2023) [23] stated that the increase in organic matter …”. In addition, the citation style “Name et al.” is not preferable in this journal with the numbering citation style like [23].

3-4) The chapter “4. Discussion” is superficial. Deeper discussion is needed.

3-5) The paragraph consisting only of one sentence in the chapter “5. Conclusion” is not preferable. Connect a few paragraphs to make a longer paragraph. In addition, the conclusion is too general to conclude something. The authors can focus more on their own results for the presence of Vibrio.

Minor points:

1. (Abstract) Results show that Vibrio spp. Persist year-round … > The results show that Vibrio spp. were detected year-round …

2. (Abstract) Please write out the full spelling of CTX without abbreviating it, when it appears at the first time.

3. Figure 3 can be deleted. The sampling locations can be plotted in figure 1. In addition, both figure 5 and figure 6 are unnecessary.

4. In the section 2.5 Molecular Identification, The exact product name of PROMEGA commercial kit must be mentioned with the name and location of the company.

5.The captions of table 4 and table 5 must include the name of target gene, because the PCR is targeted not to Vibrio spp. or Vibrio cholerae but to zinc metalloproteinase gene or to cholera toxin gene.

6.  The discussion chapter is poorly written.

Comments on the Quality of English Language

The quality of the language is below the standard of international journals. There are many typing errors. The discussion chapter is poorly written. In addition, the scientific presentation has to be improved.

Author Response

Response to Reviewer 2 Comments

Dear Reviewer 2,

Thank you very much for taking the time to review this manuscript. Your observations and recommendations, have been helpful in refining our research and enhancing the overall quality of the manuscript. Please find the detailed responses below and the corresponding corrections highlighted in the color red. Please see the attachment.

Comments 1:  The quantity of the results is not enough for the hull article. Only the presence of two genes was presented without quality control. Response 1: Thank you for pointing this out. We understand your concerns regarding the number of results presented in the article. As a result, we have thoroughly reviewed the manuscript and identified critical points. We have made the necessary corrections and implemented a more rigorous quality control process in handling the information to provide a more comprehensive and robust view of the study. This change can be found in the section "2.5. Molecular identification," page 4, paragraph 1, lines 132 to 135. “2.6. Statistical analysis”, page 5-6, lines 167 to 175. 3.3. “Molecular Identification”, page 7, lines 228 to 233; page 9, lines 266 to 270.

 To this end, the objective was restructured, this change can be found in the section "Introduction," page 2, paragraph 3, lines 70 to 73. I appreciate your feedback, as it has been instrumental in improving the manuscript.

Comments 2:  The quality of the language is below the standard of international journals. There are many typing errors. Response 2: Thank you for pointing this out. We agree with this comment. With the aim of avoiding the observed typing errors, the manuscript has been thoroughly reviewed and the identified errors have been corrected. Additionally, the manuscript has been revised in terms of presentation style and was improved by a translator experienced in the language.

Comments 3: The scientific presentation skill by the authors is low. Response 3: The journal's author guidelines were reviewed, the manuscript was revised and corrected, and it was formatted appropriately to enhance its scientific presentation. Additionally, the manuscript has been revised in terms of presentation style and was improved by a translator experienced in the language.

Comments 3-1: Section “3.1 Water and Land Use” can be moved to the materials and methods section, because section 3.1 is a part of the description of the target watershed. Response 3-1: Thank you very much for the suggestion, the information remains in the results section, since it is information gathered from interviews with cooperative members, Key informants, etc. this change can be found in the section “2.2. Identification of Pollution Sources” page 3, paragraph 1, lines 100 to 105. However, a change was made to the title of the section for better understanding. This change can be found in the section “3.1. Identification and description of point sources in the study area” page 6, lines 178.

Comments 3-2: In the section of “3.2 Physicochemical parameters of water”, the authors can prepare a figure or a table to present their measurements. Response 3-2: Thank you for pointing this out. We agree with this comment. Therefore, we have created a table for the best representation of the physicochemical parameters of water. This change can be found in the section “3.2. Physicochemical parameters of water” page 7, lines 224 to 226.

Comments 3-3: Past tense must be used for the citation in the text. For example, “Amaral et al. (2023) [23] state that the increase in organic matter …” > “Amaral et al. (2023) [23] stated that the increase in organic matter …”. In addition, the citation style “Name et al.” is not preferable in this journal with the numbering citation style like [23]. Response 3-3: Thank you for pointing this out. We agree with this comment. The “Discussion” section was revised and modified according to the guidelines of the authors' guide for the citation of references. This change can be found in the section “4. Discussion”, page 9, line 268 to 377.

Comments 3-4: The chapter “4. Discussion” is superficial. Deeper discussion is needed. Response 3-4: We agree with this comment. Therefore, we worked on the discussion of the results of our research with those of other authors. This change can be found in the section “4. Discussion”, paragraph 1, page 9, line 273 to 289; paragraph 3, 297 to 300; page 9-10, paragraph 4, 308 to 320.

Comments 3-5: The paragraph consisting only of one sentence in the chapter “5. Conclusion” is not preferable. Connect a few paragraphs to make a longer paragraph. In addition, the conclusion is too general to conclude something. The authors can focus more on their own results for the presence of Vibrio. Response 3-5: We agree with this comment. Therefore, we restructured the conclusions section according to the results obtained linked to the research objective and placed it in the suggested format. This change can be found in the section “5. Conclusions” page 11, line 381 to 395.

Minor points:

Comments 1: (Abstract) Results show that Vibrio sppPersist year-round … > The results show that Vibrio sppwere detected year-round. Response 1: The summary section was restructured and modified for better understanding. Your comments and suggestions were considered, and therefore, it was explained as follows: The HA gene was identified in the three seasons of the year, while the CTX gene was only present in the dry and rainy seasons. This change can be found in the section “Abstract” page 1, line 18 to 20.

Comments 2: (Abstract) Please write out the full spelling of CTX without abbreviating it, when it appears at the first time. Response 2: Thank you very much for your recommendation, what was suggested was carried out. In addition, in each section when mentioned for the first time, the full name was placed before the abbreviation. This change can be found in the section “Abstract” page 1, line 12 to 14.

Comments 3: Figure 3 can be deleted. The sampling locations can be plotted in figure 1. In addition, both figure 5 and figure 6 are unnecessar. Response 3: Agree. We have, accordingly, modified the map of the study area to locate the study sites. This change can be found in the section “2.1. Study area” page 3, line 88. In addition, figures 5 and 6 were eliminated.

Comments 4: In the section 2.5 Molecular Identification, the exact product name of PROMEGA commercial kit must be mentioned with the name and location of the company. Response 4: The suggested change was made, it now reads “using the Wizard® Genomic DNA Purification kit (Promega Corporation; Madison, WI USA)” This change can be found in the section “2.5. Molecular Identification” page 4, line 132 to 133.

Comments 5: The captions of table 4 and table 5 must include the name of target gene, because the PCR is targeted not to Vibrio spp. or Vibrio cholerae but to zinc metalloproteinase gene or to cholera toxin gene. Response 5: Agree. We have, accordingly changed it now reads in the table 4 “Amplification conditions for the detection of HA gene of Vibrio sp.” and table 5 “. Amplification conditions for the detection of the CTX gene of Vibrio cholerae”. This change can be found in the section “2.5. Molecular identification” page 4, line 154 and 156.

Comments 6: The discussion chapter is poorly written. Response 6: We agree with this comment. The “Discussion” section was revised and modified according to the guidelines of the authors' guide for the citation of references. This change can be found in the section “4. Discussion”, page 9, line 268 to 377.

Comments on the Quality of English Language:

The quality of the language is below the standard of international journals. There are many typing errors. The discussion chapter is poorly written. In addition, the scientific presentation has to be improved. Response: The manuscript has been revised in terms of presentation style and was improved by a translator experienced in the language.

Round 2

Reviewer 1 Report

Comments and Suggestions for Authors

The article manusccript has been corrected. It is now written in clear logic language and represents an important study. The authors identified the presence of zinc metalloproteinase (HA) gene specific for Vibrio sp. and CTX toxin gene specific for Vibrio cholerae in relation to various sources of pollution caused by anthropogenic activities carried out in the coastal zone of Manding. These sources were identified by them in a preliminary reconnaissance study and thus logically selected 10 representative sites for the study. Modern molecular techniques were applied. The article is ready for publication. I support this study.

Please make small corrections at next stage of paper preparation:

Vibrio cholerae is a pathogenic enterobacterium from the genus Vibrio which leads to a gastroenteric syndrome. As a result, the presence of these bacteria in bodies of water with varying salinity could act as a reservoir for the spread of virulent strains, posing an epidemic risk [10].

Lines 62-66

Therefore, bacteria in water bodies with varying salinity can act as a reservoir for virulent strains, posing a potential epidemic risk [10]. One method for detecting the cholera toxin gene in V. cholerae is molecular analysis of Vibrio strains collected from aquatic environments.

Line 72 …. associated with various pollution sources due to anthropogenic activities carried out in in the Mandinga Coastal… 

Reviewer 2 Report

Comments and Suggestions for Authors

The authors revised the manuscript. This reviewer doe ot have further comment for revision.